# The Effectiveness of Indoor Residual Spraying for Malaria Control in Sub-Saharan Africa: A Systematic Protocol Review and Meta-Analysis

**DOI:** 10.3390/ijerph22060822

**Published:** 2025-05-23

**Authors:** Moses Ocan, Kevin Ouma Ojiambo, Loyce Nakalembe, Geofrey Kinalwa, Alison A. Kinengyere, Sam Nsobya, Emmanuel Arinaitwe, Henry Mawejje

**Affiliations:** 1Africa Center for Systematic Reviews and Knowledge Translation, Makerere University College of Health Sciences, Kampala P.O. Box 7072, Uganda; ojambok@gmail.com (K.O.O.); nakaloy2011@gmail.com (L.N.); alison.kine@gmail.com (A.A.K.); 2Department of Pharmacology, School of Biomedical Sciences, College of Health Sciences, Makerere University, Kampala P.O. Box 7072, Uganda; geofkinalwa3@gmail.com; 3Clinical Epidemiology Unit, Department of Medicine, College of Health Sciences, Makerere University, Kampala P.O. Box 7072, Uganda; 4Department of Pharmacology, College of Health Sciences, Soroti University, Arapai P.O. Box 211, Uganda; 5Department of Pharmacology, Fins Medical University, Fort Portal P.O. Box 909, Uganda; 6Albert Cook Library, College of Health Sciences, Makerere University, Kampala P.O. Box 7072, Uganda; 7Infectious Disease Research Collaboration (IDRC), Kampala P.O. Box 7475, Uganda; samnsobya@yahoo.co.uk (S.N.); earinaitwe@idrc-uganda.org (E.A.); mawejjehenry@yahoo.com (H.M.)

**Keywords:** indoor residual spraying, insecticides, malaria, vector control, sub-Saharan Africa

## Abstract

Background: Indoor residual spraying (IRS) is a core insecticide-based vector control tool employed in most malaria-affected settings globally. However, mosquito vectors have developed resistance to nearly all of the insecticides currently used in IRS. This has necessitated a transition to new classes of insecticides, from mostly using dichlorodiphenyltrichloroethane (DDT) and pyrethroids from 1997 to 2010 to carbamates in 2011 and organophosphates in 2013. In addition, other vector control measures, like the use of long-lasting insecticide-treated bed nets (LLINs), have also been employed for malaria control. Despite the implementation of these mosquito vector control interventions, malaria remains a disease of public health concern, especially in sub-Saharan Africa, which bears over 90% of the disease burden. This review will thus collate evidence on the effectiveness of IRS for malaria control in sub-Saharan Africa. Methods and analysis: The systematic review will be conducted following a priori criteria developed using the PRISMA guidelines. Articles will be obtained through a search of the Web of Science, Google Scholar, Medline via PubMed, Scopus and Embase databases. Mesh terms and Boolean operators (“AND”, “OR”) will be used in the article search. Additionally, websites of malaria research institutions will be searched. The article search will be conducted by two independent librarians (AAK and RS). All identified articles will be transferred to EPPI-reviewer v6.15.1.0 software. Article screening and data abstraction will be performed in duplicate by four reviewers (KOO, LN, GK and MO), and any further disagreements will be resolved through discussion and consensus. We shall extract data on the country, region, study design, insecticide combination, season, susceptibility procedure used, vector control interventions, population, mosquito species, malaria incidence or prevalence, insecticide efficacy, susceptibility, genotypic resistance, vector mortality and knockdown effect. Data analysis will be performed using STATA *v*17.0. Effect sizes will be statistically pooled using inverse-variance-weighted random-effects meta-analysis. Heterogeneity and publication bias in the articles will be assessed using the I^2^ statistic and a funnel plot, respectively. For the studies that will not be included in the meta-analysis, a narrative synthesis will be written following the Cochrane Consumer and Communication Review Group format. Results: The findings of this review will help generate evidence on the effectiveness of indoor residual spraying using WHO pre-qualified insecticides in malaria control in sub-Saharan Africa. This protocol was registered in PROSPERO, registration number CRD42024517119.

## 1. Introduction

An estimated 263 million cases of malaria were reported worldwide in 2023, with an incidence of 60.4 cases per 1000 at-risk people [1,2]. This represents an increase in the incidence from 58.6 cases per 1000 people at risk in 2022, and an increase of 11 million cases from the year before. With an expected 94% and 95% of all malaria cases globally in 2023 occurring in the WHO African Region, the region continues to bear the greatest burden of the illness [1,2]. Between 2021 and 2023, fatalities dropped to 59.7 per 100,000 people at risk, while between 2000 and 2023, the malaria fatality rate declined from 28.5 to 13.7 per 100,000 people at risk [3], with children under the age of five accounting for over 76% of all malaria deaths. Indoor residual spraying (IRS) is one of the current WHO-recommended insecticide-based vector control strategies which are intended to reduce and, ultimately, interrupt malaria transmission [4,5]. According to the WHO, IRS is the application of a long-lasting, residual insecticide to potential malaria vector resting surfaces, such as internal walls, eaves and ceilings, of all houses or structures [4,5]. Indoor residual spraying aims at (i) reducing the vector’s lifespan to less than the time it takes for the malaria sporozoites to develop, which is approximately between 10 days and 4 weeks [6], depending on the plasmodium species and temperature; (ii) reducing the vector density by immediate killing; and (iii) reducing the human–vector contact through a repellent effect, thereby reducing the number of mosquitoes that enter the sprayed rooms [4,5].

Although an estimated 2.2 billion cases of malaria and 12.7 million fatalities have been prevented since 2000, according to recent WHO figures, the disease is still a major worldwide health concern, especially in the WHO African Region [1,2]. The WHO African Region alone prevented 1.7 billion cases and 12 million deaths from malaria between 2000 and 2023 [1,2]. Between 2020 and 2023, approximately 177 and 118 million people were protected by IRS globally and in the WHO Africa region, respectively [1,2,4,5]. The indoor residual spraying campaigns in most malaria-affected areas are supported by the United States Agency for International Development (USAID)/President’s Malaria Initiative (PMI) [7], and therefore, given the recent executive order by the President of the United States to halt all foreign aid and disband USAID, these efforts are bound to face serious setbacks. IRS implementation targets high-transmission settings in malaria-affected regions [2].

Some of the commonly recommended classes of insecticides used as active ingredients for IRS include pyrethroids (such as alpha-cypermethrin, deltamethrin and lambda-cyhalothrin), organophosphates (e.g., malathion, pirimiphos-methyl), carbamates (i.e., bendiocarb and propoxur), neonicotinoids, such as clothianidin, and organochlorines (e.g., dichlorodiphenyltrichloroethane, DDT) [8]. These insecticide classes have different residual activities, costs and efficacies in the field [9]. Among these, pyrethroids are the most frequently used insecticides due to their relatively low toxicity to humans, fast knockdown effect and cost-effectiveness [8,10,11].

IRS is, however, affected by high costs, complex implementation logistics and community acceptance [12]. Additionally, the emergence of resistance to most insecticides used for IRS has also been reported in most sub-Saharan African (SSA) countries [13]. The resistance of malaria vectors to pyrethroids and other classes of insecticides is widespread in sub-Saharan Africa [11,13]. Resistance to IRS is driven by the selection pressure placed on resistance genes, heavy reliance on one class of insecticides for vector control and continued use of the same chemical classes as agricultural pesticides [8].

Several strategies have been used to delay resistance to the already existing insecticides, including regularly changing insecticides (rotations), the use of a combination of insecticides with different modes of action, the deployment of insecticides of different modes of action in neighboring geographical areas (mosaic spraying) and the co-deployment of different interventions in the same place [14]. Combination insecticide mixtures have the dual potential to improve malaria vector control in addition to managing resistance [15]. One of the recent WHO-approved insecticide mixtures is a formulation of a wettable powder product containing 500 g/kg of clothianidin and 62.5 g/kg of deltamethrin [15].

Despite the widespread application of indoor residual spraying and other mosquito vector control measures, malaria remains a disease of public concern, especially in sub-Saharan Africa [2]. This is further worsened by the emerging and widespread resistance to the insecticides used in IRS and bed nets [11,16,17,18,19,20]. The introduction of interventions such as combination insecticide preparations could potentially improve the effectiveness of insecticide-based mosquito vector control interventions, including IRS [15]. Individual studies have assessed the efficacy of the insecticides used in IRS in most malaria-affected regions; however, with the malaria prevalence remaining high in sub-Saharan Africa, the effectiveness of IRS in malaria control remains unknown [15,21,22,23]. This effectiveness systematic review and meta-analysis will collate evidence on the effectiveness of the use of WHO pre-qualified insecticides in indoor residual spraying for malaria control in sub-Saharan Africa. This information will help inform the policy and decision-making in selecting insecticide compounds or combination preparations to be used in different malaria-affected geographical locations in sub-Saharan Africa.

### 1.1. Rationale

Insecticide-based interventions have contributed to approximately 78% of the reduction in the malaria burden in sub-Saharan Africa since 2000, and are the main malaria vector control measure [1,2,22,23,24]. However, the implementation of IRS is currently affected by the emergence and spread of insecticide resistance [16,18,20,24]. This resistance is mainly driven by increased selection pressure and the use of the same insecticides in agriculture. The emergence of insecticide resistance in *Anopheles* mosquitoes in sub-Saharan Africa has implications for vector control interventions. This has led to a transition to different classes of insecticides that are used in IRS for mosquito vector control [16,18,20,24]. The introduction of Anopheles stephensi in Africa in 2014 is also forcing the malaria community to re-evaluate prevention strategies, as this vector thrives in towns and cities by breeding in man-made water containers and is a primary vector of urban malaria [25].

Several reviews have been recently conducted to synthesize evidence on the effectiveness of IRS on malaria transmission; however, these have several limitations [21,26,27]. All of these reviews focused on the effect of IRS in reducing the malaria burden or prevalence as the primary outcome, which is not the only indicator of IRS effectiveness [21,26,27]. A review by Giming et al. (2023) focused only on reactive IRS application in the control of malaria [26]. However, malaria remains prevalent in most countries in sub-Saharan Africa, which commonly use proactive IRS applications [26]. Another systematic review by Pryce et al. (2022) focused on the effect on malaria of additionally implementing IRS using non-pyrethroid-like or pyrethroid-like insecticides in communities currently using ITNs; hence, the results may not directly provide evidence on the actual effect of IRS when used alone [27]. In a review by Zhou et al. (2022) [21], pyrethroids were identified to show the greatest performance in malaria control, while Pryce et al. (2022) [27] showed that adding non-pyrethroid insecticides to bed nets had the greatest effect on the malaria burden. In summary, the review by Pryce et al. (2022) [27] did not find a significant effect of adding pyrethroid insecticides to bed nets in malaria control. Additionally, Zhou et al. (2022) included studies that used DDT, methylcarbamate and primiparous-methyl insecticides only [21]. There is thus still an evidence gap on the effectiveness of other insecticides approved by the WHO for IRS. Additionally, the WHO recently approved combination/mixed IRS, such as a combination of clothianidin-deltamethrin. However, no review has evaluated the available evidence on its impact on the malaria burden in sub-Saharan Africa. The current review therefore seeks to collate evidence on the effectiveness of WHO pre-qualified single and combination insecticides used in IRS for malaria control in sub-Saharan Africa, with a focus on the effect on the malaria burden and other vector-related outcomes such as mosquito mortality, insecticide susceptibility and knockdown rate.

### 1.2. How the Intervention Might Work

Indoor residual spraying is one of the preventive measures aimed at eliminating malaria globally. It involves the use of insecticides that reduce the number and longevity of mosquito vectors, thereby decreasing malaria transmission [26]. Compared to long-lasting insecticide nets, which provide a barrier for mosquito vectors, IRS acts robustly in different ways, including repelling mosquito vectors and reducing the malaria transmission ability of the vectors and the vector mortality [5,28]. Furthermore, the current introduction of combination insecticide preparations has improved efficacy and potentially reduced the risk of resistance [5,15]. Resistance to insecticides is the greatest risk to IRS, especially in the high-burden settings common in sub-Saharan Africa [5,10,11,18,20,24]. The introduction of new insecticide classes in addition to combination agents has contributed to recent gains in malaria control, despite the effects of vector resistance [5].

## 2. Materials and Methods

### 2.1. Protocol Registration

We shall follow the Preferred Reporting Items for Systematic Reviews and Meta- (PRISMA) Guidelines [29] to perform this systematic review. This protocol is written according to the Preferred Reporting Items for Systematic Review and Meta-Analysis Protocols (PRISMA-Ps) [30] and was registered in PROSPERO with the registration number CRD42024517119.

### 2.2. Primary Review Question

What is the effectiveness of WHO pre-qualified insecticides used in indoor residual spraying for malaria control in sub-Saharan Africa (SSA)?

Secondary Review Question (s)

What is the prevalence of genotypic resistance to the insecticides used in IRS among mosquito vectors in sub-Saharan Africa?What are the species of malaria mosquito vectors prevalent in sub-Saharan Africa?What factors are associated with the effectiveness of indoor residual spraying for malaria mosquito vector control in sub-Saharan Africa?

### 2.3. Key Elements of the Review Question (PICOST)

Population: All people living in malaria-affected settings in sub-Saharan Africa and all malaria mosquito vectors in malaria-affected settings in sub-Saharan Africa.

Intervention: Application of indoor residual spraying (IRS) for malaria control using insecticides pre-qualified by the WHO.

Comparator: No indoor residual spraying (IRS) and other mosquito vector control measures, e.g., Long-Lasting Insecticide-Treated Nets (LLINs).

Outcomes: Primary outcome—Malaria prevalence in communities following indoor residual spraying (IRS); and secondary outcomes—Knockdown rate of mosquito vectors following exposure to IRS, mosquito vector mortality rate following exposure to IRS, type and class of insecticide used in IRS, residuality of the insecticide on different wall surfaces, mosquito vector insecticide susceptibility (phenotypic resistance), mosquito vector insecticide resistance genes (molecular resistance), mosquito vector (s) (species), community acceptability and feasibility, how insecticides were used in IRS (single or combination preparations), nature of indoor residual spraying (proactive, reactive and focal IRS).

Study Designs: Randomized controlled trials (RCTs), case–control studies, cohort studies, interrupted time series, before-and-after designs and cross-sectional studies/surveys.

Timeframe: Studies conducted from the year 1990 to 2025.

### 2.4. Eligibility Criteria

#### 2.4.1. Inclusion

Articles that report on the impact of IRS performed using WHO pre-qualified insecticides.Articles published in peer-reviewed journals.Articles published in all languages (no language restriction).

#### 2.4.2. Exclusion

Articles that do not segregate the effects of multiple mosquito vector control measures on the malaria burden in sub-Saharan African countries.Articles that report on vector insecticide susceptibility outside of the context of IRS.Articles that did not obtain and report ethical review and approval.

### 2.5. Identification of Primary Studies

An experienced librarian (AAK) and the principal investigator (OM) will independently search for articles from four established databases. The articles from the two independent searches will then be merged in EndNote software and the duplicates removed.

### 2.6. Information Sources

Articles published in peer-reviewed journals reporting on the effectiveness of indoor residual spraying (IRS) vector intervention for malaria control will be searched from the Web of Science, Google Scholar, MEDLINE via PubMed, Scopus and Embase databases. The search will cover a period from the year 1990 to 2025 and will include sub-Saharan African countries. Furthermore, we will conduct a hand search on institutional websites for any relevant grey literature. We will also screen through the reference lists of the included studies and relevant systematic and literature reviews for additional eligible articles.

### 2.7. Search Strategy

The scoping literature search was finished on 18 November 2024; however, the full article search has not yet been carried out. To find relevant articles based on PICOST, the search terms listed below will be used in the full article search. The Boolean operator “OR” will be used to combine terms that relate to the same PICOST element, while “AND” will be used to join terms that relate to separate concepts or PICO categories.

### 2.8. Search Terms

The following search terms will be used; ‘deltamethrin’, ‘bendiocarb’, ‘primiphos-methyl’, ‘DDT’, ‘dichlorodiphenyl-trichloroethane’, ‘malathion’, ‘temephos’, ‘fenitrothion’, ‘cypermethrin’, ‘chlothianidin’, ‘insecticide’, ‘Actellic’, ‘chlorfenapyr’, ‘propoxur’, ‘pyrethroid’, ‘neonicotinoid’, ‘Sumishield’, ‘Anopheles gambiae s.l’, ‘Anopheles funestus’, ‘Anopheles arabiensis’, ‘Anopheles stephensi’, ‘Anopheles pharoensis’, ’Ace1’, ‘cytochrome P450s’, ‘glutathione S-transferases’, ‘nAChR target site’, ‘Cyp9k1’ and ‘Tep’, ‘mosquito’, ‘malaria vector mosquito’, ‘mosquitoes’, ‘susceptibility’, ‘efficacy’, ‘sensitivity’, ‘knockdown’, ‘mortality’, ‘delayed mortality’, ‘residuality’, ‘residual life’, ‘indoor residual spraying’, ‘IRS’, ‘spray technique’, ‘malaria transmission’, ‘season’, ‘rainfall season’, ‘rebound malaria epidemics’, ‘malaria epidemics’, ‘insecticide resistance’, ‘insecticide tolerance’, ‘resistance’, ‘resistance genes’, ‘molecular marker’, ‘resistance alleles’ and ‘sub-Saharan Africa’. The search string will be developed using the above terms.

### 2.9. Data Management and Study Selection

For the initial management of references from the search results, EndNote *v*20 software will be used. The articles will then be exported to EPPI-reviewer v6.15.1.0 software. The articles will then be screened in duplicate using the predetermined eligibility criteria. The screening will be performed independently in duplicate by the review team (MO, GK, KOO and LN) in EPPI-Reviewer v6.15.1.0, using a screening tool developed a priori, and piloted using 10% of the search yield. Kappa agreement [31,32] of 80% will be used and any disagreements between the reviewers resolved through discussion and consensus. Any further disagreements will be referred to the tiebreaker (MO).

### 2.10. Data Abstraction and Coding

The data abstraction tool will be created in Microsoft Excel spreadsheet 2007 and piloted using 10% of the eligible studies. The final tool will then be uploaded in EPPI-Reviewer v6.15.1.0. The coding process will be carried out independently in pairs by research team members (MO, GK, KOO and LN). Kappa agreement [31,32] of 80% will be used and any disagreements resolved through discussion and consensus. The data will later be validated for quality control by an independent senior reviewer (OM) to ensure completeness and correctness.

### 2.11. Data Items

The following data categories will be abstracted: administrative information (author, year of publication, DOI, country/region, funding source), methods (study design, population, sample size) and results (insecticide efficacy, resistance, knockdown effect, susceptibility) (Table 1).

### 2.12. Outcomes and Prioritization

#### 2.12.1. Dependent Variables

Prevalence of malaria in communities following indoor residual spraying (IRS) using insecticides for mosquito vector control in sub-Saharan Africa.Mosquito vector knockdown effect.Residuality (residual efficacy).

Mosquito vector insecticide molecular resistance genes.

#### 2.12.2. Independent Variables

Types of mosquito vectorsInsecticides used in indoor residual spraying for mosquito vector control in sub-Saharan Africa.Types of insecticides (single or combination compound).Factors associated with efficacy of insecticides in indoor residual spraying in sub-Saharan Africa (SSA).

### 2.13. Risk of Bias Assessment

Two research team members (MO, GK, LN and KOO) will independently assess the methodological quality of the included observational studies using a modified version of the Newcastle–Ottawa Scale (NOS) [33]. The tool includes seven domains rated from 0 (high risk of bias) to 3 (low risk of bias); the mean of the domains is considered to result in a score between 0 and 3, with a higher score indicating a lower risk of bias. Consensus on any disagreements in the quality assessment will be reached after discussion and consultation with an independent senior reviewer.

For randomized controlled trials (RCTs) and non-randomized trials of interventions, we will use the Cochrane Risk of Bias tool and the Risk of Bias in Non-Randomized Studies—of Interventions (ROBINS-I) tool to assess the potential risk of bias [34,35,36]. The bias is measured as a rating (high, low or unclear) for individual elements from the five domains (selection bias, attrition bias, performance bias, reporting bias, detection bias and other biases such as conflict of interest). In addition, the tools provide for the assessment of concerns for the applicability of the study to the systematic review, which will also be assessed. Kappa agreement [31,32] of 80% will be used and any disagreement resolved through discussion and consensus. Any further unresolved disagreements will be referred to the tiebreaker (MO).

### 2.14. Publication Bias

The included publications will be assessed for publication bias using the asymmetry of funnel plots and the Egger test as applicable [37,38]. These are rank-based data augmentation strategies that are effective at detecting publication bias caused by missing data or studies. We will generate funnel plots and utilize their symmetry to assess the possibility of publication bias among the articles included in the review. In the absence of missing studies, the scatter plot resembles a symmetrical inverted funnel with a broad base and a narrow top. The plot’s huge “holes” and asymmetry suggest publication bias; however, this could also be explained by other causes, such as study heterogeneity.

### 2.15. Assessment of Strength and Confidence of Cumulative Evidence

A modified GRADE approach will be used to evaluate the overall strength of the evidence. The approach has eight ratings that can be assigned as follows; 1A: strong recommendation of high-quality evidence, 2A: weak recommendation of high-quality evidence, 1B: strong recommendation of moderate-quality evidence, 2B: weak recommendation of moderate-quality evidence, 1C: strong recommendation of low-quality evidence, 2C: weak recommendation of low-quality evidence, 1D: strong recommendation of very low-quality evidence and 2D: weak recommendation of very low-quality evidence. We will assign the certainty of the evidence ratings to the aforementioned outcome elements using a method developed by the GRADE Working Group [39], and we will perform this in duplicate. Any disagreements will be resolved through consensus.

### 2.16. Heterogeneity

The I^2^ statistic will be used to determine the degree of statistical heterogeneity in the included articles. The I^2^ statistic will display the percentage (%) of heterogeneity due to between-study variation [40,41,42]. The heterogeneity will be classified as low (I^2^ = 25%), moderate (I^2^ = 50%) or high (I^2^ > 75%). A subgroup analysis will be performed on articles with low to moderate heterogeneity [43].

### 2.17. Criteria for Determination of Independent Findings

Dependency can occur at the study or intra-study levels. In the case of numerous reports from the same study, the most thorough and recent report, when available, will be used. However, an integrative strategy will be utilized to treat the data from all of these reports as a single case if they span several subgroups or outcomes [44]. Each meta-analysis will only include one effect at the intra-study level from every single study. Studies that show more than one effect for different outcome types will be synthesized separately. Before incorporating the “synthetic effects” in a meta-analysis, we will utilize them to calculate a sample-weighted average in cases where studies report several dependent effects for a particular outcome type.

#### 2.17.1. Missing Data

If the published articles include missing data, the study authors will be notified. When the author cannot be contacted or does not respond, we will list the study’s characteristics, but exclude it from the meta-analysis.

#### 2.17.2. Data Analysis and Synthesis

The data analysis for this review will be performed using STATA v17. Standardized mean differences (SMDs) for the continuous outcome variables and odds ratios (ORs) or prevalence ratios (PRs) for the dichotomous outcome variables will be analyzed separately. The effect sizes will be statistically pooled using an inverse-variance-weighted random-effects meta-analysis [45,46]. The random-effects model will be used to calculate the pooled mean effect size, as the effect size is likely to vary between the different studies [46]. Additionally, the random-effects models will enable statistical inferences to be made from a population of studies other than those included in the meta-analysis [47].

The pooled effects will be stated using an appropriate metric, such as a percentage change in the odds or a mean difference in the natural units of the outcome. In studies where several effect sizes are reported from the same sample, the average of the combined effect sizes will be computed. In studies with overlapping samples, an overall estimate will be calculated, and those that report effect sizes from distinct subgroups will be treated as separate samples in the meta-analysis. The synthesis will be provided as a summary of the findings tables, basic graphs and forest plots. This will adhere to the format of the Cochrane Consumer and Communication Review Group [48]. We will summarize the included articles, categorize them based on the study design and type of intervention, organize and tabulate the results to identify patterns and convert the results into a standard descriptive format. These will take the shape of result data tables, simple graphs or forest plots as needed. These will be included in the summary of the findings tables, which will help to guide the synthesis for dissemination. We will employ both narrative and quantitative syntheses.

### 2.18. Sensitivity Analysis

The sensitivity analysis will be performed by eliminating studies from the meta-analysis one at a time to see if the meta-analysis results are sensitive to any particular study [49]. We will also look into the sensitivity of the findings to different levels of bias.

## 3. Results

The review will generate findings on the effectiveness of indoor residual spraying for mosquito vector control. The prevalence of malaria in communities following the implementation of the IRS will be reported in this review. Additionally, we shall report on the extent of the resistance to WHO pre-qualified insecticides, including combination preparations, used in IRS for mosquito vector control in sub-Saharan Africa. The prevalence of the molecular markers of insecticide resistance among mosquito vectors in SSA will also be reported. We shall also report the factors associated with the effectiveness of indoor residual spraying (IRS) mosquito vector intervention for malaria control in SSA.

## 4. Discussion

This study will systematically collate the evidence available on the effectiveness of WHO pre-qualified insecticides used in indoor residual spraying for malaria mosquito vector control in sub-Saharan Africa. Despite the long-term use of IRS for malaria control, there is limited information on the effectiveness of this vector intervention, especially in the presence of other interventions such as LLINs. By collating information about the knockdown effect, susceptibility, residuality (residual efficacy) and moderating factors like the types of mosquito vectors; the molecular resistance among malaria mosquito vectors; the types of insecticides used for IRS; and other factors associated with the efficacy of the insecticides used in indoor residual spraying in SSA, the findings from this study will provide guidance on the selection of insecticides for use in IRS. This is critical, especially due to the current stalling of malaria eradication efforts in most malaria-affected countries.

## 5. Amendments

This protocol may be amended in the event that issues arise during the review process; if such changes are made, they will be reported in the review article and published as deviations from the protocol.

## 6. Conclusions

The effectiveness of mosquito vector control interventions including IRS and LLINs is threatened by the reported emergence and spread of insecticide resistance. Consequently, this is the case as evidenced by the current resurgence of malaria, especially in sub-Saharan Africa, despite the implementation of multiple vector control interventions. There is thus a need for quality evidence to guide the selection of insecticides to use in mosquito vector control interventions such as IRS and LLINs, especially in highly malaria-burdened settings.

## Figures and Tables

**Table 1 ijerph-22-00822-t001:** Review result items/areas.

Item	Description
Administrative data	This will collect data to identify the articles, including the authors, citations, funding and countries
Method	Data will be collected on the study designs, insecticides (name, formulation and strength), combinations, seasons, susceptibility procedures used, vector control interventions, populations and mosquito species
Results	Malaria incidence, malaria prevalence, insecticide efficacy, susceptibility (phenotypic resistance), genotypic resistance, vector mortality and knockdown effect
Setting	Countries in sub-Saharan Africa

## Data Availability

All data that will be generated by this review will be contained in previously published studies, and an outline of the included studies and extracted data will be made publicly available with the results manuscript.

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
