# Peer review of "The Effectiveness of Indoor Residual Spraying for Malaria Control in Sub-Saharan Africa: A Systematic Protocol Review and Meta-Analysis"

_ijerph, 2025, doi:10.3390/ijerph22060822_

Round 1

Reviewer 1 Report

Comments and Suggestions for Authors

As attached

Author Response

Comment: I want to thank the authors for coming up with this proposal to review the effectiveness of IRS in sSA. The outcome will be important in the policy framework for malaria control. To make the work better, I have made the following suggestions

Response: Thank for your kind words and taking time to review this manuscript. We are grateful.

Comment: It is good that the authors mentioned the morbidity per 1000. It would also be good to state the morality as this shows the ultimate impact of malaria on the African population

Response: Thank you for this observation, we have provided further details on malaria burden including mortality “Between 2021 and 2023, fatalities dropped again to 59.7 per 100,000 people at risk while between 2000 and 2023, the malaria fatality rate declined from 28.5 to 13.7 per 100,000 people at risk [3] with children under the age of five accounted for over 76% of all malaria deaths.”

Comment: Line 61: According to the WHO…

Response:  Thank you for your observation we made that correction as follows “According to the WHO, IRS is the application of a long-lasting, residual insecticide to potential malaria vector resting surfaces such as internal walls, eaves and ceilings of all houses or structures”

Comment: Line 64: This clause “. time it takes for the malaria sporozoites to develop, “is not clear. The authors need to make this clearer by stating the time it takes the malaria sporozoite to develop 

Response:  Thank you for this observation. We have included the time it takes for sporozoites to develop to make sentence clearer and provided a reference for the same. “Indoor residual spraying aims at i) reducing the vector’s lifespan to less than the time it takes for the malaria sporozoites to develop which is approximately between 10 days to 4 weeks [5] depending on the plasmodium species and temperature”

Comment:  Line 75: It is good that the authors mentioned the roles played by the USAID in funding IRS. The authors need to highlight how the current executive order from the Trump US administration affects this aid. 

Response: Thank you for this observation. We highlighted anticipated impact of the executive order from the US government. “Indoor residual spraying campaigns in most malaria-affected areas are supported by the United States Agency for International Development (USAID)/President’s Malaria Initiative (PMI) [6] and therefore given the recent executive order by the president of United States to halt all foreign aid and disband USAID this effort are bound to face serious setbacks” 

Comment: Line 109: the authors should state the type of review here. 

Response:  Thank you for this comment. We have clearly stated the type of review. “This effectiveness systematic review and meta-analysis will collate evidence on the impact of the use of WHO pre-qualified insecticides in indoor residual spraying for malaria control in sub-Sahara Africa”

Comment: Line 152: “acts robustly in different “is not clear. Please make clearer 

Response: We have re-written the sentence for clarity as follows “IRS acts robustly in different ways including repelling mosquito vectors, reducing malaria transmission ability of vectors and vector mortality”

Comment: Line 148: I do not understand the essence of this subsection. Move to methods or expunge

Response: Thank you for your comment. It is very important that we demonstrate how the intervention we trying to evaluate works to the readers of who some may not be from the field of study hence this section and usually it comes immediately before the study questions hence its position.

Comment: Line 167: The primary research question does not align with the study aim “To collate evidence on the effectiveness of the use of WHO pre-qualified insecticides in indoor residual spraying for malaria control in sub-Sahara Africa.” According to the authors in the rationale for the study, “Previous systematic reviews have reported that previous studies showed the impact of IRS on malaria prevalence” The authors need to harmonize this information to be consistent. 

Response: Thank you for that observation we have re-aligned the primary research question with study aim as follows “What is the effectiveness of the WHO pre-qualified insecticides used in indoor residual spraying for malaria control in sub-Sahara Africa”

Comments: Line 182: write LLINs in full

Response: The word has been written in full as “Long Lasting Insecticide Treated Nets (LLINs)”

Comment: Line 191: The exact search dates should be indicated. For instance, from Jan, 1 1990 to Jan 1, 2025.

Response: We have made that clarification as suggested “Timeframe: Studies conducted in the year 1990 to 2025”

Comment: Line 205: Did you mean to say articles without ethical review and approval?

Response:  

Yes, we meant articles that did not obtain and report ethical review and approval. The necessary changes have been made to make it clear “Articles that did not obtain and report ethical review and approval”

Comments: The authors should also include other article types excluded like systematic and literature reviews. 

Response: Thank you for this comment. Relevant systematic and literature reviews will be used for reference searching to identify relevant primary studies for Inclusion and we have edited the information sources section to highlight this. However, this being a systematic review and meta- analysis and not a systematic review of reviews we can not include systematic reviews among eligible designs for inclusion. We will also screen through reference lists of included studies and relevant systematic and literature reviews for additional eligible articles”

Comment: Line 208: The databases should be indicated 

Response:  The four databases have been listed on line 213 and 214 “Articles published in peer-reviewed journals reporting on the effectiveness of indoor residual spraying (IRS) vector intervention for malaria control will be searched from; Web of Science, Google Scholar, and MEDLINE via PubMed, Scopus, and Embase databases”

Comment: Line 214: Indicate the exact date as suggested above 

Response:  We have indicated the exact dates as follows “The search will cover a period from the year 1990 to 2025 and will include sub-Saharan African countries”

Comment: Line 239: I presume this should be EPPI Reviewer. The authors should be consistent 

Response:  We corrected it for consistency “The articles will then be exported to EPPI-reviewer v6.15.1.0 software. The articles will then be screened in duplicate using predetermined eligibility criteria”

Comment: During the search and data extraction, how will articles not published in English be translated? This has to be clearly established whether it will be done manually or by an automated system

Response: Thank you for this comment, we shall use google translate to translate articles not published in English-to-English Language

Comment: Line 277: The authors mentioned that RCTs will be included in the article selection. That is okay. However, NOS is not suitable for this group of studies. How will the authors assess the risk of bias for RCTs? This has to be clearly stated. There are other tools that can e used to perform risk of bias assessment for RCTs. The authors can choose any of them they deem appropriate. 

Response:  

Thank you for this comment from Line 283-291 we have described how Randomised controlled trials and non randomised studies of interventions will be assessed for risk of bias as follows “For randomized controlled trials (RCTs) and non-randomized trials of interventions, we will use the Cochrane Risk of Bias (RoB.2) tool and Risk of Bias in Non-Randomized Studies - of Interventions (ROBINS-I) tool to assess the potential risk of bias”

Comment: Results: okay 

Response: Thank you for this observation 

Comment: Conclusion: Okay

Response: Thank you for this observation 

Comment:Overall comment: Minor grammar corrections 

Response: Thank you for this observation, all grammatical errors have been addressed to improve the readability of the manuscript

Reviewer 2 Report

Comments and Suggestions for Authors

Dear Authors,

Thank you very much for allowing me to review your manuscript. Your proposed study sounds very important and interesting, and I'm looking forward to reading your eventual manuscript. I do have a number of questions and suggestions about your current manuscript describing the protocol:

  1. After reading your protocol, I understand now that you're proposing a meta-analysis. However, this is not clear from your title. Please consider including this term in the title.
  2. On line 29, please don't capitalize "Dichloro..."
  3. On line 42, is the "Epi-reviewer" really "EPPI-Reviewer"?
  4. In the Methods and Analysis section of the Abstract, can you please include what data will be extracted and analyzed from the studies, and briefly describe how they will be analyzed?
  5. On line 130, please put a period after "al".
  6. On line 136, I don't quite understand why you used the term "Therefore". It implies a logical connection between the previous and the current sentence. However, just because Pryce et al. found that the addition of non-pyrethroid insecticides to bed nets had the greatest effect on malaria burden, it is not a logical consequence that they didn't find a significant effect of adding pyrethroid insecticides to bed nets. 
  7. On line 142, please don't capitalize "Clothianidin-Deltamethrin".
  8. On line 142, please start a new sentence at "however".
  9. On line 152, please change "different" to "different ways".
  10. Please end the sentence on line 165 with a period.
  11. I was surprised that your Introduction and Rationale didn't mention the introduction of invasive Anopheles stephensi into Africa given it's insecticide resistance.
  12. On line 166, I was surprised to see that your Primary Review Question is limited to the effect of IRS on prevalence of malaria. I thought that is what you criticized in the studies you mentioned in the second paragraph of page 4. What about other, for example, entomological measures of effectiveness of IRS? You mention how you're going to look at those on line 146. Why are they not part of your Primary Review Question, especially since they are listed in the Outcomes section of your Key Determinants (PICOST)?
  13. On line 172, what do you mean by "sub-Sahara Africain"?
  14. In the Outcomes section of your Key Determinants of the Review Question (PICOST), I would argue that you list some "outcomes" that are not really outcomes of the intervention, but factors that determine the outcome of the intervention. These would include: Type and class of insecticide used in IRS; Mosquito vector species; Community acceptibility and feasibility; How insecticides are used; Nature of indoor residual spraying. I would recommend moving these either into the Population or Intervention section.
  15. On line 215, what do you mean by "hand search"? Please explain.
  16. On page 7, in the section on "Search terms", the first set of search terms are not italicized, until "Sumishield", while after that they are italicized. Are those two sets of concepts of PICO categories, so there would be an "AND" between the two sets?
  17. On line 239, is the "Epi-reviewer" really "EPPI-Reviewer"?
  18. On line 243, can you provide a citation for Kappa agreement? Do you mean Cohen's Kappa?
  19. In Table 1, I recommend listing "mosquito species" in the Methods section as data that will be collected
  20. In the Outcomes and Prioritization section on page 9, I believe several of the "Independent variables" should really be "Dependent variables", such as "Mosquito vectors knockdown effect", "Residuality (residual efficacy)" and "Mosquito vector insecticide molecular resistance genes", especially since these are listed under "Results" in Table 1.
  21. On line 273, what does "SSA" stand for?
  22. I would recommend to move the "Missing data" and "Data analysis and synthesis" sections from page 10 to before the "Risk of bias assessment" section on page 8.
  23. In the "Publication bias" section on page 9, please describe that numbers will be represented on the funnel plot.
  24. In the section "Assessment of strength and confidence of cumulative evidence", please describe in more detail the modified GRADE method.
  25. The sentences from line 335 to 339, starting from "In studies where multiple effect sizes..." seem to belong to the section on "Criteria for determination of independent findings". They don't seem to be talking about the pooled effect.
  26. On line 359, please change "sub-Sahara" to "sub-Saharan". Perhaps on line 90 is where you could introduce the abbreviation "(SSA)" after the full term for the first time.
  27. Please add a line break after line 374.
  28. On line 382, please replace one of the two "especially" with some other word.
  29. In the Data Availability Statement, it would be nice if you could state that you will make the extracted data publicly available as a supplementary material.

Author Response

Comment: Thank you very much for allowing me to review your manuscript. Your proposed study sounds very important and interesting, and I'm looking forward to reading your eventual manuscript. I do have a number of questions and suggestions about your current manuscript describing the protocol

Response: Thank you for your kind word and taking time to review this protocol manuscript, we will be happy to publish the results manuscript upon completion.

Comment: After reading your protocol, I understand now that you're proposing a meta-analysis. However, this is not clear from your title

Response: 

Thank you for this observation, we have modified the study title and it reads; Effectiveness of Indoor Residual Spraying for Malaria Control in sub-Saharan Africa: A Protocol Systematic Review and Meta-Analysis

Comment: On line 29, please don't capitalize "Dichloro..."

Response: Thank you for this comment we have edited this as follows “This has necessitated a transition to new classes of insecticides from mostly using dichlorodiphenyltrichloroethane (DDT)...”

Comment: On line 42, is the "Epi-reviewer" really "EPPI-Reviewer"?

Response: Thank you for this observation we edited this throughout the manuscript for consistence. The correct spelling is EPPI-Reviewer.

Comment: In the Methods and Analysis section of the Abstract, can you please include what data will be extracted and analysed from the studies, and briefly describe how they will be analysed?

Response: Thank you for this suggestion we have edited the abstract as advised 

Comment: On line 130, please put a period after "al".

Response: Thank you for this observation this change has been made as follows “A review by Giming et al. (2023) focused only on reactive IRS application in the control of malaria”

Comment: On line 136, I don't quite understand why you used the term "Therefore". It implies a logical connection between the previous and the current sentence. However, just because Pryce et al. found that the addition of non-pyrethroid insecticides to bed nets had the greatest effect on malaria burden, it is not a logical consequence that they didn't find a significant effect of adding pyrethroid insecticides to bed nets. 

Response: Thank you for that observation we removed the word “therefore” and the sentence now reads; “In summary, the review by Pryce et al., (2022) did not find a significant effect of adding pyrethroid insecticide to bed nets in malaria control”

Comment: On line 142, please don't capitalize "Clothianidin-Deltamethrin".

Response:  Thank you for that observation we have mad the necessary changes as follows “Additionally, the WHO recently approved combination/mixed IRS such as a combination of clothianidin-deltamethrin”.

Comment: On line 142, please start a new sentence at "however".

Response:  Thank you for that observation we started a new sentence as follows “However, no review has evaluated the available evidence on their impact on malaria burden in sub-Saharan Africa. ”

Comment: On line 152, please change "different" to "different ways".

Response:  Thank you for that observation we have edited and the sentence now reads “IRS acts robustly in different ways including repelling mosquito vectors...”

Comment: Please end the sentence on line 165 with a period.

Response: Thank you for that observation we have edited accordingly.

Comment: I was surprised that your Introduction and Rationale didn't mention the introduction of invasive Anopheles stephensi into Africa given its insecticide resistance.

Response:  Thank you for this observation we have added a few lines highlighting this in the rationale as follows “The introduction of Anopheles stephensi in Africa in 2014 is also forcing the malaria community to re-evaluate the prevention strategies because vector thrives in towns and cities by breeding in man-made water containers, and is a primary vector of urban malaria”

Comment: On line 166, I was surprised to see that your Primary Review Question is limited to the effect of IRS on prevalence of malaria. I thought that is what you criticized in the studies you mentioned in the second paragraph of page 4. What about other, for example, entomological measures of effectiveness of IRS? You mention how you're going to look at those on line 146. Why are they not part of your Primary Review Question, especially since they are listed in the Outcomes section of your Key Determinants (PICOST)?

Response:  We have revised the primary review question to align it with the study aim and it reads as follows “What is the effectiveness of the WHO pre-qualified insecticides used in indoor residual spraying for malaria control in sub-Sahara Africa” We believe this question is broad enough to encompass both entomological outcomes and prevalence of malaria as measures of effectiveness.

Comment: On line 172, what do you mean by "sub-Sahara African"?

Response:  Thank you for this observation, the region sub-Saharan Africa refers to the countries that lie south of the Sahara Desert as classified by World Bank.

Comment: In the Outcomes section of your Key Determinants of the Review Question (PICOST), I would argue that you list some "outcomes" that are not really outcomes of the intervention, but factors that determine the outcome of the intervention. These would include: Type and class of insecticide used in IRS; Mosquito vector species; Community acceptability and feasibility; How insecticides are used; Nature of indoor residual spraying. I would recommend moving these either into the Population or Intervention section.

Response: Thank you for this observation. We have listed the outcomes and factors that are associated with or influence the outcome upon Indoor residual spraying. Some of these factors pertain to the intervention, the vector, population and the community.

Comment: On line 215, what do you mean by "hand search"? Please explain.

Response: I appreciate you making this observation. In order to enhance electronic database searches with possibly pertinent studies, we refer to hand searching as the process of manually going through institutional study publication webpages, page by page. Because some studies might not be indexed in electronic databases or may not be found by electronic searches because of insufficient indexing or the lack of pertinent keywords in titles or abstracts, this helps guarantee that every relevant study is identified. 

Comment: On page 7, in the section on "Search terms", the first set of search terms are not italicized, until "Sumishield", while after that they are italicized. Are those two sets of concepts of PICO categories, so there would be an "AND" between the two sets?

Response: Thank you for this observation we have italicised all search terms as it should be. This just a list of search terms without Boolean operators “AND, OR”. These terms will be combined by the Boolean operators to form a search string and it will be published with the results manuscript as an appendix.

Comment: On line 239, is the "Epi-reviewer" really "EPPI-Reviewer"?

Response: Thank you for this observation we edited this throughout the manuscript for consistence. The correct spelling is EPPI-Reviewer.

Comment: On line 243, can you provide a citation for Kappa agreement? Do you mean Cohen's Kappa?

Response: Thank you for this observation. We have provided the appropriate references as advised

Comment: In Table 1, I recommend listing "mosquito species" in the Methods section as data that will be collected

Response: We appreciate your observation we have added mosquito species under methods in Table 1

Comment: In the Outcomes and Prioritization section on page 9, I believe several of the "Independent variables" should really be "Dependent variables", such as "Mosquito vectors knockdown effect", "Residuality (residual efficacy)" and "Mosquito vector insecticide molecular resistance genes", especially since these are listed under "Results" in Table 1.

Response: We appreciate this observation we have moved mosquito vectors knockdown effect, residuality (residual efficacy) and mosquito vector insecticide molecular resistance genes to dependent variables section.

Comment: On line 273, what does "SSA" stand for?

Response: Thank you this comment SSA stand for sub-Sahara Africa and we have written it in full form

Comment: I would recommend to move the "Missing data" and "Data analysis and synthesis" sections from page 10 to before the "Risk of bias assessment" section on page 8.

Response: Thank you for this comment. We have written following the PRISMA-P guidelines and also bearing in mind the sequence of events during the systematic review and believe the Risk of Bias assessment comes before data analysis and synthesis and hence this order in the manuscript

Comment: In the "Publication bias" section on page 9, please describe that numbers will be represented on the funnel plot.

Response: We appreciate your comment and suggestion however, we don’t quite get it clearly to be able to paraphrase this section. We understand a funnel plot as a graphical tool that depicts the effect size as a function of its standard error. The two slanting lines of the funnel represent the confidence intervals and the dots within the funnel represent the individual studies. The plot, especially the bottom part, will be symmetric when the publication bias is absent (Borenstein et al., 2009). We believe we have sufficiently described the above in this this section.

Comment: In the section "Assessment of strength and confidence of cumulative evidence", please describe in more detail the modified GRADE method.

Response: Thank you for this suggestion. We have provided a more detailed description for GRADE ratings as guided.

Comment: The sentences from line 335 to 339, starting from "In studies where multiple effect sizes..." seem to belong to the section on "Criteria for determination of independent findings". They don't seem to be talking about the pooled effect.

Response: We appreciate your comment. The description starting with line 335 to 339 shades light on how studies with multiple effect sizes and sub-group analysis which do not report a single overall effect size will handle in the during meta-analysis or pooling of effect sizes from different studies. Hence, we believe its appropriate to keep it where it is, thank you!

Comment: On line 359, please change "sub-Sahara" to "sub-Saharan". Perhaps on line 90 is where you could introduce the abbreviation "(SSA)" after the full term for the first time.

Response: Thank you for this observation we made the necessary changes as advised.

Comment: Please add a line break after line 374.

Response: Thank you for that observation we made the necessary edit as advised.

Comment: On line 382, please replace one of the two "especially" with some other word.

Response: We have made that edit as advised the sentence now reads “Consequently, this the case as evidenced by the current resurgence of malaria especially in sub-Saharan Africa despite the implementation of multiple vector control interventions”

Comment: In the Data Availability Statement, it would be nice if you could state that you will make the extracted data publicly available as a supplementary material.

Response:  Thank you for this suggestion we have included it and now it reads as follows “All data that will be generated by this review will be contained in already published studies and an outline of included studies and extracted data will be made publicly available as supplementary material with the results manuscript. ”